# Disorder of consciousness rather than complete Locked-In Syndrome for end stage Amyotrophic Lateral Sclerosis: a case series
Florent Gobert [1,2,12] ✉, Inès Merida[3,12], Emmanuel Maby[4,12], Perrine Seguin [4], Julien Jung [4,5], Dominique Morlet[4], Nathalie André-Obadia[5], Frédéric Dailler[1], Christian Berthomier [6], Anatole Otman[4], Didier Le Bars[3,7], Christian Scheiber[8], Alexander Hammers [9,10], Emilien Bernard [11], Nicolas Costes[3], Romain Bouet[4] & Jérémie Mattout [4]

## Abstract

**Background** The end-stage of amyotrophic lateral sclerosis (ALS) is commonly regarded as a complete Locked-In Syndrome (cLIS). Shifting the perspective from cLIS (assumed consciousness) to Cognitive Motor Dissociation (potentially demonstrable consciousness), we aimed to assess the preservation of covert awareness (internally preserved but externally inaccessible) using a multimodal battery.

**Methods** We evaluate two end-stage ALS patients using neurophysiological testing, passive and active auditory oddball paradigms, an auditory Brain-Computer Interface (BCI), functional activation-task imaging, long-term EEG, brain morphology, and resting-state metabolism to characterize underlying brain function.

**Results** Patient 1 initially follows simple commands but fails twice at BCI control. At follow-up, command following is no longer observed and his oddball cognitive responses disappear. Patient 2, at a single evaluation, is unable to follow commands or control the BCI. Both patients exhibit altered wakefulness, brain atrophy, and a global cortico–subcortical hypometabolism pattern consistent with a disorder of consciousness, regarded as an extreme manifestation of ALS-associated fronto-temporal dementia.

**Conclusions** Although it is not possible to firmly prove the absence of awareness, each independent measure concurred with suggesting that a "degenerative disorder of consciousness" rather than a cLIS may constitute the final stage of ALS. This condition appears pathophysiologically distinct from typical tetraplegia and anarthria, in which behavioural communication and BCI use persist to enhance quality of life. Identifying the neuroimaging signatures of this condition represents a substantial milestone in understanding end-stage ALS. Large-scale longitudinal investigations are warranted to determine the prevalence of this profile among patients whose communication appears impossible.

## Plain language summary

Some patients with amyotrophic lateral sclerosis become progressively paralyzed to the extent that they can no longer communicate. This raises a critical question: does consciousness remain despite a complete loss of interaction? To address this issue, we analysed brain activity in two patients at the late stage of this disease. Our findings indicate that no reliable signs of awareness could be detected, even when employing advanced methods to assess brain function, including approaches designed to enable communication without movement. Two hypotheses may account for this outcome: either prolonged paralysis progressively leads to the disappearance of consciousness, or the degeneration of brain function itself directly underlies its loss. However, our conclusions ultimately depend on the reliability of consciousness assessment tools, even when used in combination.

Amyotrophic lateral sclerosis [ALS] is a neurodegenerative disorder characterized by a progressive motor weakness with muscle atrophy leading to tetraplegia and respiratory distress, with death occurring 2 to 3 years from onset. ALS is associated with frontotemporal dementia (FTD) in 15% of patients, but cognitive and behavioural impairments occur in up to 80% by end-stage disease[1]. Extra-motor extension of ALS-related abnormalities has previously been described for brain metabolism[2,3], neurotransmitter function[4,5], or perfusion[6] at the population level, but are rarely prescribed for diagnosis purposes at the individual level. Morphological biomarkers of initial ALS without cognitive impairment are focused on corticospinal tract

involvement[7]. Abnormal encephalic [$^{18}$F]fluoro-deoxy-glucose [FDG] Positron Emission Tomography [PET] has been proposed as a fairly accurate and early diagnosis tool in ALS[8,9], while cortical morphometry is normal at the early stage[10]. In case of early cognitive impairment, frontal FDG-PET hypometabolism could be associated with motor and somato-sensory cortical atrophy, in line with the diagnosis of ALS-FTD[2]. In case of ALS without cognitive or behavioural symptoms, the atrophy could be observed rather in the brainstem[10].

As riluzole and non-invasive ventilation are[11] currently the only therapies extending survival[12], symptomatic treatments are focused on improving the quality of life and delaying the loss of autonomy[13]. Therefore, continuing an eye-based communication (saccades or blink) after speech disappearance is critical. Saccades are classically spared in early ALS[14], but subclinical saccadic disorders have been detected as an early sign of a frontal decline[15]. Beyond the point of saccade-based communication disappearance, the functional state associated with the ultimate stages of ALS has been poorly investigated but clinical reports of late symptoms suggested a large diffusion of the neurodegenerative pathology involving sphincterian and complex oculomotor disorders[16]. Histological data confirmed widespread loss of pyramidal neurons in the frontal cortex and anterior cingulate areas[17]. In most cases, this last stage is not reached because tracheotomy is not proposed, refused, or an unexpected lethal complication occurs earlier. If chosen, tracheostomy, by extending survival to an average of 7 years[18], might also increase the probability of reaching these extreme stages of the disease[19]. Proposing and performing tracheostomy is highly dependent on the healthcare system and cultural background, varying from 5% in Europe[20] to 30% in Japan[21].

In rare instances, the final cognitive status cannot be behaviourally measured because of a complete motor failure that forbids any inter-individual communication. At this point, the clinical picture in this degenerative process resembles an entity which, in the post-acute brain literature has been described as "Locked-In Syndrome", presenting at least minimal facial/ocular movements after severe brainstem vascular injury[22]. The concept of functional[23] or complete[24] Locked-In Syndrome [cLIS] (that could be included in the recent concept of "cognitive-motor dissociation"[25,26]) was then introduced. It hypothesizes that paraclinical tools could assess cognitive or consciousness contents in spite of the inability to behaviourally demonstrate them[26,27]. In the ALS context, trying to communicate with cLIS patients by non-motor ways is of utmost importance for practical and ethical reasons[28]: to adapt treatment to pain and to confirm the patients' will to carry on supportive care in the absence of daily communication. This motivated the development of brain-computer interfaces dedicated to the restoration of these patients' communication[29]. In a comprehensive description of prolonged (seven-year) BCI use[30], device function was limited to calling caregivers, the communication itself relying more importantly on the last available motor response. As the ultimate failure of BCI use in this case was contemporary with the loss of reliable motor responses, even this longest longitudinal description of BCI in ALS did not involve a case of cLIS. The most functionally versatile BCI to date[31] was able to substantially improve communication ability for a still communicating patient at the stage of tetraplegia and dysarthria. Altogether, BCI tools have not been proven useful in the cLIS context so far[32].

Some authors tried to define whether some consciousness dimensions could be impaired in the ALS-related cLIS context but neither poly-somnographic sleep-wake patterns[33], EEG spectrum[34] not EEG complexity[32] were found sufficiently altered so far to conclude definitely on the existence of "disorders of consciousness" [DOC]. However, simple brain responses usually tested in DOC (Somatosensory Evoked Potentials[35]) were found absent or altered in other cohorts, indicating the heterogeneity of the end-stage ALS population and the possibility of more severe stages of the neurodegenerative process. For instance, the intra-cerebral neural signal could change gradually over years, with lower amplitude of EEG changes during mental motor imaging and sensory stimulations[30].

In the present study, we adapted a "multimodal awareness assessment battery" [MAAB] to a neurodegenerative context, after a cLIS-like

progressive loss of saccade control among two ALS patients with prolonged survival after their tracheotomy. For a comprehensive phenotypic description of their consciousness level, wakefulness fluctuations, and brain metabolism were investigated as well.

These two ALS patients with severe motor impairment fail to demonstrate any sign of communication (no use of the BCI) and eventually show no reliable evidence of awareness (one lost previously observed command-following and cognitive responses, while the other never exhibited them) and an impairment of wakefulness. Neuroimaging reveals brain atrophy and global cortico–subcortical hypometabolism consistent with a "degenerative disorder of consciousness", that appears to be distinct from cLIS as communication should have remained possible. These findings suggest that end-stage ALS evolves towards a unique pathophysiological condition marked, long after the occurrence of the complete-LIS stage, by a progressive and degenerative loss of consciousness. However, this interpretation undoubtedly remains constrained by the limits of current assessment tools.

## Methods
### Inclusion & Ethics
As the patient management was modified to answer a clinically relevant question (communication ability by paraclinical tools) and not for research purposes, this individualised analysis was conducted in accordance with the law on data protection (version no. 2004-801, 6 August 2004). The results of neurophysiological and imaging explorations were given to the physician in charge of patients' treatment. In addition, informed consents to transfer the patients to the teaching hospital for the MAAB were obtained before starting the exploration from legal representatives.

The time course of each assessment and its relation to the medical history is summarised in Fig. 1.

Authorization was obtained from the ethics committee of the Hospices Civils de Lyon (Comité Scientifique et Éthique des Hospices Civils de LYON CSE-HCL – IRB 00013204; Pr Cyrille Confavreux; approval N. 24-310).

### Electrophysiological assessments
**EEG recordings.** EEG data were recorded according to a routine clinical protocol, using a Micromed® EEG recording system (11-electrode EEG for 20 min and 15-electrode monitoring system for 24 h). EEG data were then interpreted by two trained neurophysiologists (NAO and FG). The long-term EEG was analysed using the ASEEGA algorithm of automatic sleep classification (CB, Physip®) using the Cz-Pz derivation and trained on normative data from healthy subjects[36]. It is illustrated by the data of a healthy subject.

**Evoked potential protocol.** Standard EP data were recorded i) for Patient 1 at $T_1$ and for Patient 2 with a 7-electrode Micromed® EEG recording system (Fz, Cz, Pz, F3, F4, M1, M2) and ii) for Patient 1 at $T_2$ with a 32-electrode BrainAmp® EEG system. The reference electrode was placed on the tip of the nose, and the ground electrode on the forehead. One bipolar electro-oculogramm (EOG) derivation was recorded from 2 electrodes placed on the supra-orbital and infra-orbital ridges of the right eye. The signal was amplified (band-pass 0.3–100 Hz), digitized (sampling frequency 1000 Hz) and stored for off-line analysis.

Data from two auditory paradigms were obtained at each MAAB. The *passive oddball paradigm* was based on short-duration deviant tones (occurrence frequency of 0.14) and included the patient's own first name as a rare stimulus ("novel" with an occurrence frequency of 0.03). This protocol aimed at evidencing sensory auditory processing (N1 to standard tones), automatic change detection (mismatch negativity, MMN detected in the difference "deviants minus standards") and attention orienting (novelty P3 in the response to novels). The patient was presented with one block of 2000 stimuli, including 280 deviants and 60 instances of his own name[37]. The *active-passive oddball paradigm* (Act-Pass) relied on frequency deviant tones ($p = 0.15$) and aimed at evidencing voluntary processes[38]. The auditory stimulation was presented in two successive conditions, the patient being

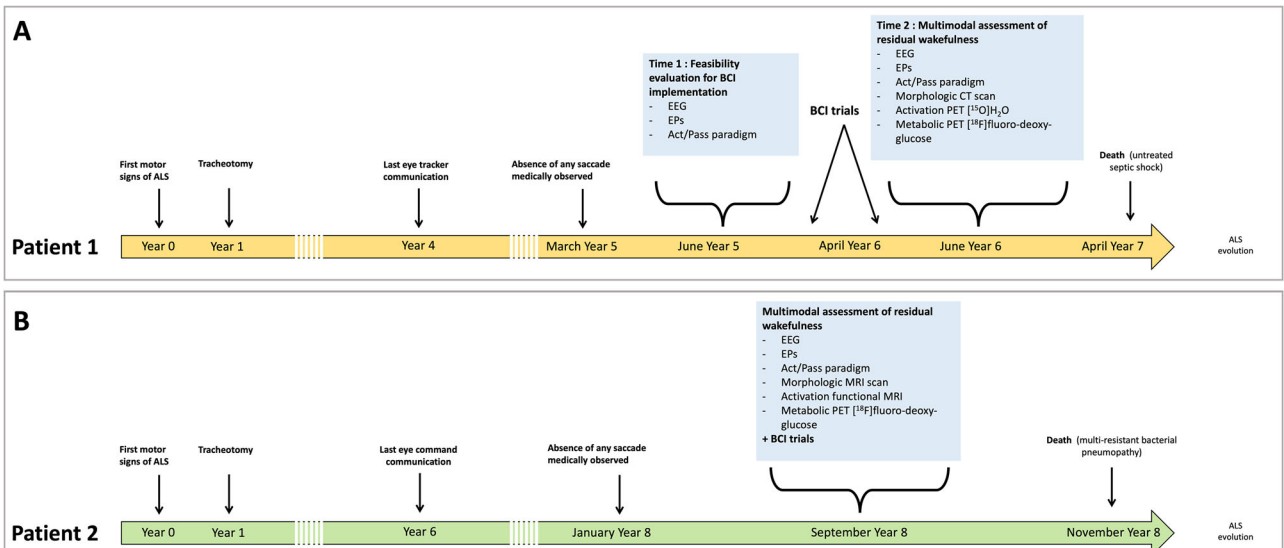

**Fig. 1 | *Study design in relation to the medical history*. A** For Patient 1, the first part of the MAAB (Time 1, $T_1$) was performed 5 years after onset and confirmed the feasibility of communicating via BCI. Two further attempts at BCI-based communication were performed at the patient's bedside 6 years after onset and failed to establish a reliable communication. During the BCI sessions, an abnormally low frequency EEG activity was observed with an inconstant reactivity to nociceptive stimulus. The patient was admitted 2 months later (Time 2, $T_2$) for another MAAB and a complementary imaging evaluation (CT scan, activation-task PET and FDG PET). **B** – For Patient 2, a single MAAB was planned 8 years after onset, together with the attempt of BCI-based communication and the complementary evaluation (MRI scan, activation-task functional MRI and FDG PET).

first asked to actively divert his attention (using a mental imagery task) and then to actively focus his attention (counting the deviants). In each condition, the patient was presented with 3 blocks of 200 stimuli, including 30 deviants, each block being preceded by recorded instructions. Voluntary processes were evidenced both by comparing responses to deviants and targets in the condition of focused-attention ("Count") and by comparing responses to deviants in the focused-attention and in the diverted-attention conditions ("Focused *vs* Diverted").

Wave detection was achieved in a two-step process, including visual identification by a trained neurophysiologist[38] and objective confirmation by means of randomization methods applied to the significant time-clusters. A one sample T-test was calculated between the two conditions (sample by sample; $p$-value < 0.05). A time-cluster was defined when many consecutive samples were significant. A cluster permutation[39] was used to solve the multiple comparisons problem, incorporate biophysically motivated constraints in the test statistic and control the false alarm rate. According to this nonparametric method, we build a permutation histogram based on 1000 random partitions of our two conditions and clusters were rejected if their statistics were above threshold (unilateral $p$-value < 0.05).

### Auditory-based Brain Computer Interface

We developed a dedicated auditory Brain Computer Interface (BCI) based on spatial selective attention[40] as standard visual BCI paradigms were not suitable in this case of complete oculomotor failure[41]. This auditory-based BCI was designed as a communication device to elicit a single reliable yes-or-no answer at a time: two interleaved trains of stimuli are presented, and the BCI is driven by the difference in responses between all attended and all unattended stimuli.

We used spoken words pronounced by a synthesized male voice ("yes" and "no") delivered in a pseudo-random order. This paradigm was benchmarked with 18 control subjects and 7 patients with severe motor disability. Seventeen out of 18 healthy subjects obtained online BCI accuracy above chance level. Results were more mitigated amongst patients as two patients with advanced ALS were at chance level, while the other two could control the BCI with perfect accuracy[41]. In contrast, the three patients with locked-in syndrome due to a brainstem injury could not control the BCI. A similar paradigm was developed and tested in Ref. 42, reporting accuracies above chance level for both patients with ALS that tried the BCI.

To answer a question, 20 standard stimuli (of 100 ms duration) and 3 deviant stimuli (of 150 ms duration) were delivered with a random Stimulus Onset Asynchrony between 400 ms and 700 ms. The patients had to orient their attention to the right side in order to answer "Yes" and to the left in order to answer "No".

EEG referenced to the nose was recorded from 22 active electrodes (Acticap system-Brain Products, Germany) at positions Fp1, Fp2, F7, F3, Fz, F4, F8, FC5, FC1, FC2, FC6, C3, Cz, C4, CP5, CP1, CP2, CP6, P7, P3, P4, P8 following the extended 10-10 system placement[43].

All parameters learned from the calibration session were subsequently applied during the test sessions. These parameters pertained to the feature selection and subsequent classification steps. The former consisted in spatial filters derived from the xDAWN algorithm[44]. Then a two-class Naive Bayes classifier was trained, based on the spatially filtered training data. More information on this classifier can be found in Ref. 41. At the end of calibration, a leave-one-out based cross-validation was used to evaluate the quality of this calibration. If cross-validation was above chance level, we performed the online testing of the BCI. On the contrary, if cross-validation did not exceed chance level, we started a new calibration. The course of each patient's sessions is summarized in Fig. 2.

For the test session, the EEG stream was processed in real-time by a framework developed in Python: a band-pass filter between 0.5 and 20 Hz was applied, 500 ms long epochs were extracted to analyse the responses evoked by standard "yes" or "no" sounds.

Averaging was performed for each of these two conditions: standard "yes", standard "no". The Bayesian classifier computed the posterior probability for each class given the observed features ("yes" or "no" target stimuli versus "yes" or "no" non-target stimuli). Then the output of each classifier was optimally combined to obtain a final and unique posterior probability for each of the two classes ("yes" or "no") (see the appendix of Ref. 41). Complementary offline analyses were performed to assess BCI performance based on different stimulus types (standard, deviant, yes, no).

Patient 1 underwent two BCI sessions one week apart (6 years after ALS onset, 10 months after the $T_1$ initial MAAB, 2 months before the $T_2$

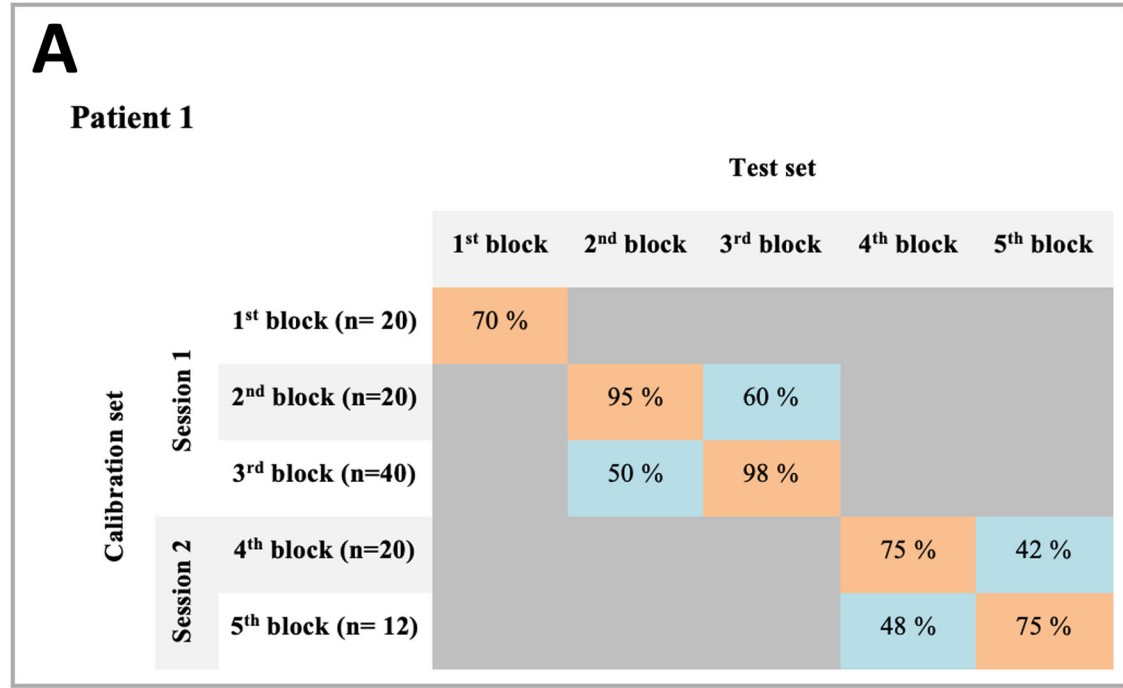

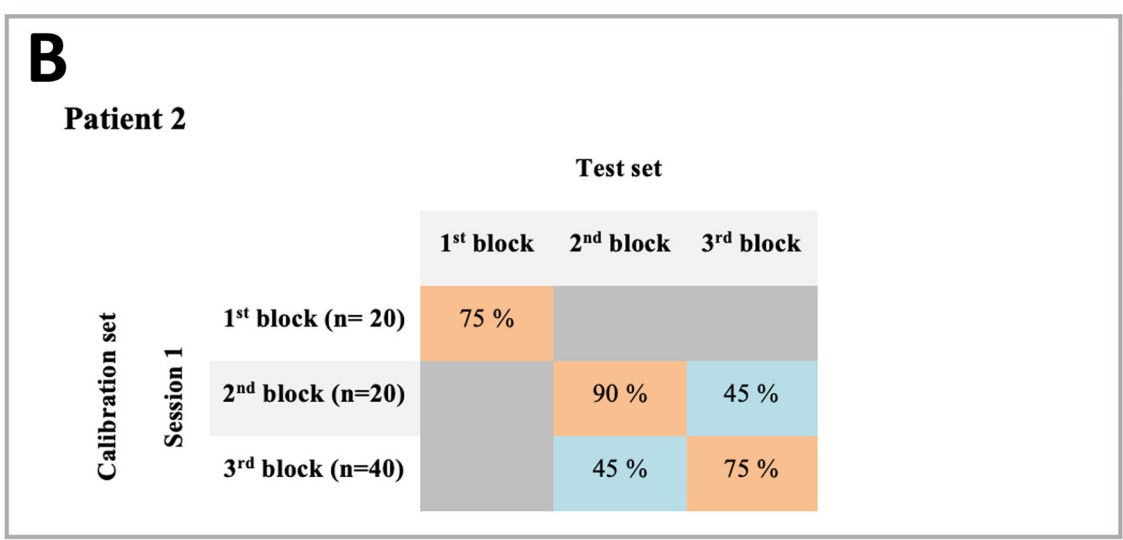

*Session: blocks performed on the same day (n= number of trials)*
*Orange: Cross-validation accuracy; Blue: Test accuracy*

**Fig. 2 | Neurophysiological data - BCI paradigm: experimental design and classification results for each patient. A** Online and offline results indicate the absence of BCI control by patient 1. **B** Online and offline results indicate the absence of BCI control by patient 2. The high cross-validation accuracy is consistent with overfitting. For $p > 0.05$, chance levels are: at 62.5 % for 40 trials, at 70 % for 20 trials, at 83 % for 12 trials.

complete MAAB). In the first one, patient 1 was asked to follow 20 instructions in the calibration session and 40 questions on basic general knowledge for the test session. For the second session, in an attempt to make the task more accessible, we decided to slow down the stimulation trains with a random Stimulus Onset Asynchrony between 600 ms and 900 ms.

Patient 2 underwent only one BCI session (8 years after ALS onset, and concomitantly to the other components of the MAAB) with an adapted (monaural) protocol because of a left peripheral deafness (as also described in ref. 41 for one patient).

### PET and fMRI Study

*Acquisition of images.* Three-dimensional PET scans were acquired on a PET-CT tomograph for Patient 1 (Biograph mCT/S 64 scanner), along with a morphologic CT scan to assess brain atrophy. For Patient 2, the protocol was adapted to the use of a PET-MRI scanner (Biograph mMR) with anatomical and functional MRI.

For both patients, FDG-PET imaging consisted of an injection of 130 MBq of [$^{18}$F]fluoro-deoxy-glucose followed by the acquisition of a static image from 40 to 50 minutes post-injection.

For the activation-task protocol and in accordance with previous studies using fMRI in non-degenerative coma[45], each patient was asked to mentally visualize himself while sequentially visiting each room in his house. For Patient 1 (PET-CT), the activation-task paradigm was designed as the changes of regional cerebral blood flow (rCBF) at rest and during the visuo-spatial mental imaging. The contrast was estimated by recording the distribution of radio-activity after two repeated intravenous injections of 300 MBq of [$^{15}$O]H$_2$O. For

Patient 2 (PET-MR), the acquisition was adapted to an fMRI protocol closely related to the original paradigm[27] ([45 s rest; 45 s task] x 10 blocks).

**Metabolic FDG-PET data analysis**. For FDG-PET, a voxel-based analysis was conducted in comparison to a database of 37 healthy subjects[46] using SPM 12 (Wellcome Department for Cognitive Neuroscience, London, UK) with Matlab R2017a (Mathworks Inc., Natick, MA®)[46]. Image processing consisted in: i) spatial normalisation to Montreal Neurological Institute (MNI) space for both patients (for Patient 1 in absence of MRI: a temporary explicit mask was defined to allow normalisation by thresholding the patient's mean uptake value at 2000 Bq/cc to remove relatively high extra-cerebral FDG uptake for the PET data; for Patient 2 the T1 MPRAGE MRI was used for spatial normalisation); ii) common smoothing (kernel size [8 8 8] mm); iii) conversion of PET images to standardized uptake value [SUV] units to reduce weight- and activity-related variability within the database.

Two statistical parametric map analyses were performed at the voxel level to objectively detect global and focal changes of metabolism, applied to each patient separately.

An analysis of variance with factor group (patient vs. healthy controls), age as covariate, and proportional scaling to global intracerebral mean SUV was used to assess the global hypometabolism visually observed by the nuclear medicine physician and to search for any voxel with a preserved metabolism in the fronto-parietal associative cortex[45]. Age was defined as covariate across each patient and the control group to reduce the effect of age-related brain atrophy. A liberal threshold was used for this analysis (uncorrected $p < 0.05$), according to the methodology proposed in ref. 45.

A second analysis of variance with group factor and co-variables "age" and "global mean value" (extracted from every SUV image within the intracranial volume mask from SPM) was performed to specify the patterns of focal metabolic changes corrected for global activity (i.e., relative hypometabolism or relative hypermetabolism in case of global hypometabolism). A more stringent threshold was used for this analysis (uncorrected $p < 0.01$) to confirm the specificity of local clusters, if any.

**Activation [¹⁵O]H₂O PET and fMRI data analysis**. The analysis was focused on parahippocampal areas, under the assumption that they would be activated by the spatial navigation task[27].

For patient 1 (PET-CT), the activation results from the paired [¹⁵O]H₂O PET scans were analysed using a SISCOM (Subtraction Ictal SPECT Co-registered to MRI) protocol[47] to allow for a statistical analysis based on voxel-by-voxel differences between the two single images. For patient 2 (PET-MR), the contrast between BOLD imaging (task–rest) was processed in SPM 12 using a general linear model contrasting periods of active imagery with periods of rest[27]. A $p$-value of <0.05 (with family-wise error rate – FWE – correction for multiple comparisons) was considered to be significant to reliably demonstrate the presence of an activation pattern.

**Statistics and reproducibility**

Statistical analyses of the data were conducted at the individual level to answer clinical questions. The results are qualitatively discussed at the phenotypic description level. The reproducibility of experiments is suggested by the application of a similar exploration on two patients at different times and would be reinforced by the extrapolation of the same protocol in different centres.

**Reporting summary**

Further information on research design is available in the Nature Portfolio Reporting Summary linked to this article.

# Results

## Population description

At the date of the first recording, Patient 1 (male, 56–60 years old) and Patient 2 (male, 70–75 years old) presented a 5- and 8-year evolving ALS,

respectively. Communication was lost 3 and 9 months earlier, respectively (Fig. 1).

Patient 1 was able to communicate using an eye-tracking system until Year 4. Saccades slowed down progressively, but eye movements could still be recognised by his relatives. In March of Year 5, the saccadic amplitude was too weak to be reliably perceived by a trained neurologist. The final complete assessment ($T_2$) took place 13 months after the loss of communication by saccadic eye movements.

Patient 2 was able to choose from a list until Year 6 and to have a reliable "Yes-No" code until the end of Year 7. Responding to simple commands by saccades was still possible until January of Year 8 after strong verbal stimulations by a speech therapist.

In both cases, after having excluded confounding local factors of saccade-based communication inability by an ophthalmologic examination, a cLIS was hypothesised despite their cognitive ability to communicate being unknown.

The behavioural classification of consciousness based on motor responses (i.e., Coma Recovery Scale-Revised [CRS-R][48]) was inconclusive to demonstrate any cortical function for both patients (CRS-R score = 2, with fixed eye-opening without stimulation related to a bilateral facial palsy). Patient 1 presented no eye-movement at all, but Patient 2 had a spontaneous and transient gaze deviation to the right that could be regarded as a pathological ping-pong gaze with a limited amplitude on the left[49]. Neither reliable voluntary control of movement nor visual-guided saccades was possible to the right.

### *Absence of control of an auditory BCI (*Fig. 2*)*

For both sessions of Patient 1 and the single session of Patient 2, the online test accuracies were at chance level.

In contrast, cross-validation accuracies within sessions were above chance level (see Fig. 2), which typically reflects overfitting.

### *A disappearance of responsiveness confirmed by passive Evoked Potentials and active processes (*Fig. 3*)*

For Patient 1, at $T_1$ the N100 response and the MMN response to duration deviant and the novelty P3 response to the patient's own name were clearly identified. At $T_2$, the novelty P3 was not identified but the N100 and MMN response to deviants persisted. For Patient 2, the neurophysiological pattern was comparable to Patient 1 at $T_2$ (excepted the left peripheral deafness leading to adapt the BCI protocol) without passive Subject Own Name [SON]-P300 responses.

For Patient 1 at $T_1$, the active paradigm showed a significant and diffuse P3b response to targets in the focused-attention condition ("Count"). These components were not found when the attention was diverted from the auditory stimuli by the mental imagery task. A significant and diffuse N2 response could be observed when comparing the responses to deviants in the focused-attention and diverted-attention conditions ("Focused *vs* Diverted"). No specific responses could be detected for Patient 1 at $T_2$.

For Patient 2, no response was observed in the "Count" condition. A significant P3 response was observed for the "Focused *vs* Diverted" condition but it was rejected as inconsistent during the clinical review process as it appeared on a single derivation (Pz) and with a late latency (750 ms).

### *An alteration of wakefulness confirmed by long-term EEG (*Fig. 4*)*

For Patient 1 at $T_1$, the standard EEG showed a physiological background activity with Alpha rhythm in posterior regions with spontaneous fluctuations. Background EEG activity was reactive to external stimulations such as light, touch, and sound (Supplementary Fig. 1). At $T_2$, the EEG suggested fluctuations of vigilance with an alternating pattern with spontaneous diffuse theta activity alternating with delta activity on a long-term recording. A prolonged period of "abnormal wake" was distinguished from two periods with increased Delta power during the night, suggesting the persistence of slow-wave sleep. This wakefulness pattern was different from the normal arousal pattern of a healthy subject and from the spectral analysis on

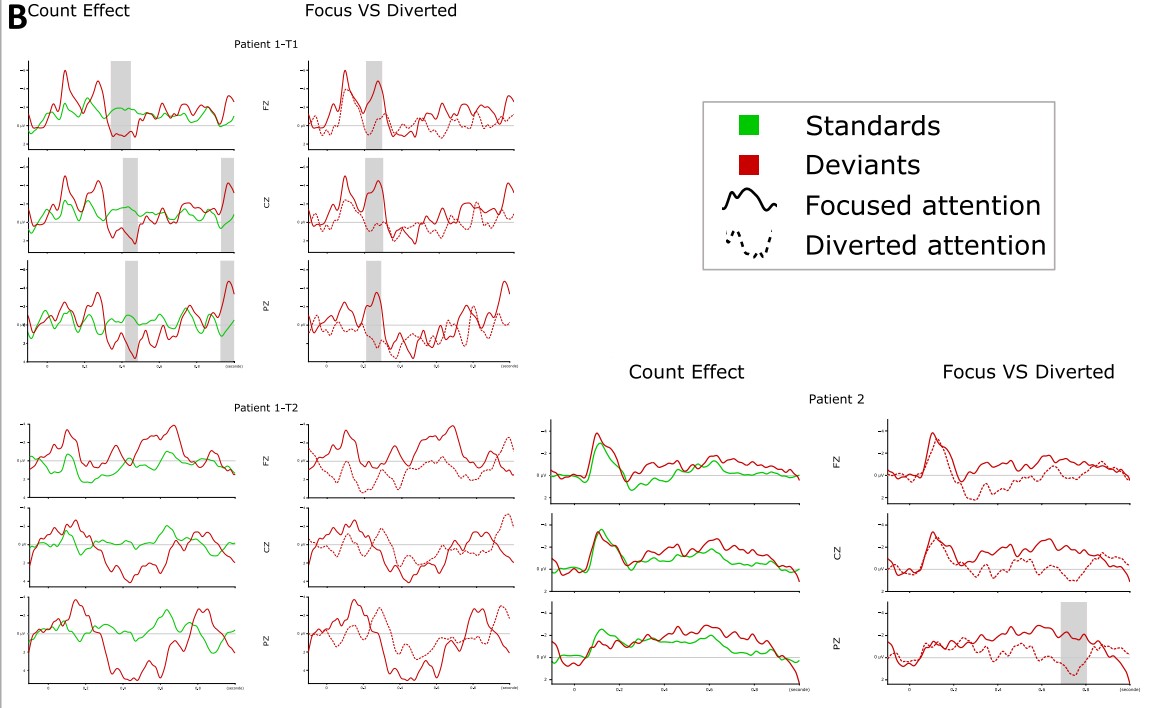

| Identity | EEG | SEPs | BAEPs | MLAEPs | N100 | MMN | SON-P300 | Act-Pass |
|---|---|---|---|---|---|---|---|---|
| Patient 1 – $T_1$ | Reactivity present | NA | Normal brainstem conduction | Normal sub-cortical and cortical conduction | + | + | + | "Count": P3b<br>"Focused Vs Diverted": N2 |
| Patient 1 – $T_2$ | Inconsistent | Normal amplitude | Normal brainstem conduction (Reduced peripheral response) | Normal sub-cortical and cortical conduction with ear-asymmetry (L<R) | + | + | 0 | "Count": No<br>"Focused Vs Diverted": No |
| Patient 2 | Reactivity present | Reduced amplitude | Normal brainstem conduction (Left peripheral deafness) | Normal sub-cortical and cortical conduction | + | + | 0 | "Count": No<br>"Focused Vs Diverted": Inconsistent |

**Fig. 3 | Neurophysiological data: Classic analysis and Active - Passive paradigm.**
**A** Summary table for neurophysiological results. **B** Active ERP paradigms: For the "Count", the deviant response is shown in red and the standard response in green. For Patient 1 at $T_1$, the comparison between red lines and green lines shows the presence of a statistically significant P3b (grey bars). For the "Focused *vs* Diverted", the focused-attention response is shown as a continuous red line and the diverted-attention response as a dotted red line. The comparison between continuous and dotted red lines shows the presence of a statistically significant N2 for Patient 1 at $T_1$ and an inconsistent late parietal P3 for Patient 2. Grey shaded bars indicate the latencies presenting statistically significant differences. SEPs Somatosensory Evoked Potentials, BAEPs Brainstem Auditory Evoked Potentials, MLAEPs Middle Latency Brainstem Auditory Evoked Potentials

20 minutes at $T_1$. The sleep classification by ASEEGA algorithm[36,50,51] (Physip®) was compatible with N2-N3 periods during the period of increased Delta power, but the mixed Theta-Alpha periods were instable and alternatively classified as N1, REM sleep or wake.

For Patient 2, brain activity was depressed but labile with few physiological sleep features. A slight reactivity of background EEG rhythm to sound and touch remained present (Supplementary Fig. 1). The sleep/wake architecture was disrupted despite sleep-like patterns occurring preferentially during the night (e.g. numerous N2 and N3 periods between 10 PM to 6 AM). However, N3 occurred also during the day (e.g. between 4:45 PM to 6PM) and prolonged arousal periods were observed during the night (e.g. around midnight).

### Brain imaging provided no further argument of active processing but could explain the overall pattern by severe atrophy and hypometabolism

The morphological scans (Fig. 5, top) demonstrated substantial brain atrophy, either in the frontal and temporal lobes (Patient 1) or more diffusely (Patient 2).

FDG-PET hypometabolism (Fig. 5, middle) was global (including in the thalamus and caudate nucleus) for both patients. According to the analysis of variance corrected for age and global metabolism, no cortical region showed relative hypometabolism or hypermetabolism. The

significant clusters of relative hypometabolism were rejected because of their extra-cerebral location in Supplementary Fig. 2, top: left lateral sulcus for Patient 1, lateral ventricles for Patient 2). The significant clusters of relative hypermetabolism (Supplementary Fig. 2, bottom) were in both cases in the periventricular white matter. This was related to the scaling with global mean correction, creating a statistical artefact as white matter brain areas were relatively less hypometabolic for both patients compared to the grey matter (mostly involved in the global hypometabolism). In other terms, the hypometabolism of white matter (whose absolute reduction of metabolism had a lower amplitude than the grey matter absolute reduction of metabolism) appeared as pseudo-hypermetabolic. This indirectly reinforced the argument for the cortical hypometabolism.

These combined analyses indicated that no voxel had a preserved metabolism within the fronto-parietal cortex because: this cortex was mostly involved in the global hypometabolism (Fig. 4, middle); neither global (Fig. 4, bottom) nor relative (Supplementary Fig. 2, bottom) hypermetabolism were found in this region.

Neither cerebral blood flow changes in [$^{15}$O]$H_2O$-PET-CT (Patient 1 – $T_2$) nor BOLD modulation in functional MRI [fMRI] (Patient 2) showed any difference in parahippocampal areas during the mental imagery task (spatial navigation[27,52]) when comparing active and passive scans (Supplementary Fig. 3).

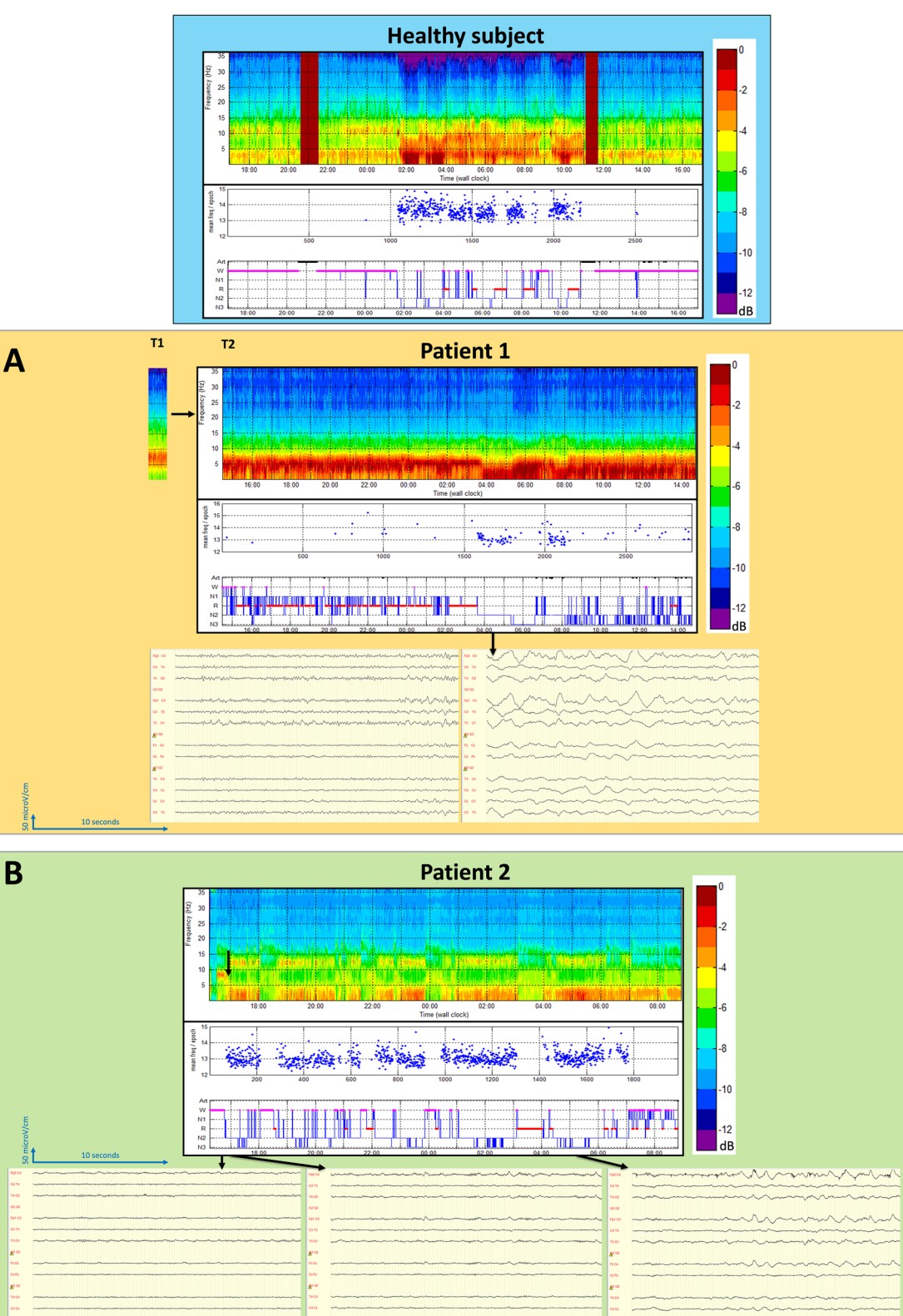

**Fig. 4 | Neurophysiological data: background EEG activity.** Selected results from ASEEGA algorithm (Physip®). On the x-axis: time of the polysomnography. On the y-axis, from top to bottom: time-frequency analysis of EEG (normalized power spectral density in dB), mean frequency of spindles per 30-seconds epoch, 5-class hypnogram. Results are presented for a healthy subject then for each patient. An illustration by raw EEG pages is provided for the most relevant patterns. **A** Patient 1 at $T_2$: Abnormal wakefulness pattern with a pathological wake EEG on the left and an N3 period on the right (Delta dominant frequency corresponding to a slow wave sleep period). On the time-frequency analysis, the increase of Theta-Delta frequency during the prolonged wake period at $T_2$ can be compared to the time-frequency analysis made on a 20-min EEG at $T_1$ (predominant Alpha-Theta frequency without Delta rhythms) and to the pattern of the healthy subject. **B** Patient 2: Abnormal sleep-wake architecture with: i) prolonged periods of wake during the evening (on the left) and the night (around midnight for instance); ii) prolonged sleep periods during the day (N2 period in the middle) and the night (N3 period on the right). EEG bipolar montage, pages of 20 seconds, filters = 0.053 – 60 Hz, amplitude = 50 microV/cm. W = Wake N1-2-3 = Sleep stage 1-2-3 R = REM sleep Art: Artefacts

## Discussion

Cases of cLIS related to severe ALS have been described multiple times in numerous case reports. However, as we discuss thereafter, most authors do not consider the hypothesis of a consciousness disorder as they strongly assume that all these patients could regain communicative abilities, provided they are given the appropriate BCI tool, which simply needs to be developed and optimised. The present demonstration suggests that this remains a strong hypothesis and that alternative explanations should be considered to ensure all aspects of the problem are addressed. Even in the most comprehensive descriptions of ALS-related cLIS[53], it appeared that there was currently no discussion in literature regarding the complete extinction of consciousness in some ALS cases, supported by an objective mixed behavioural and neurophysiological multimodal approach. We proposed to investigate the pathophysiological mechanisms underlying this disappearance using morphological and functional imaging, despite their technical difficulties[53].

Several specialised centres have proposed BCIs to communicate with patients in end-stage ALS either to improve and facilitate communication (tetraplegia and dysarthria[31]) or to call for attention before using a persistent motor function (LIS[30]) or to create a new communication device (cLIS[54]). In most cases, they did not achieve consistent on-line communication[32,55] but in some cases, succeeded[54], at least for some years before the cLIS stage[30]. Technical[55] and theoretical[56] limitations have been emphasised so far to explain failures, putting the discussion concerning the impact of pathology on the back burner[55]. Few neuroimaging data were previously available to explore the central evolution during this final global neurodegenerative condition. The recent longitudinal multi-year description of BCI use observed a dynamical and global brain atrophy on CT scans[30], but no functional exploration was provided to explore the extension of cortical dysfunction within atrophic regions and beyond.

In general, a relationship with a cognitive decline was either not considered[55] or only hypothesised[30] (as well as neural changes in the sensorimotor cortex and sustained attention) but was not nosologically associated with a DOC-like condition. In one case[57], this lack of performance was related to a probable alteration of wakefulness through prolonged 24 h EEG monitoring but without reaching the frontier with a DOC-like more radical interpretation (assuming that finding short windows of optimal wakefulness would be a universal answer). Ultimately, the authors remained focused on optimising a BCI tool to address motor difficulties by methodological advancement in EEG signal processing and to adapt their method to fluctuations of vigilance by long duration of recordings to select the moments of higher arousal.

Our multimodal explorations shed some light on this alternative explanation: BCI failed in some late-ALS patients – previously communicating by saccades – not only because eye-movements vanish, attentional processes decrease or the algorithmic interfaces were sub-optimal. In some cases, the most probable cause could be an evolving degenerative DOC. This state should be putatively considered as an alternative endpoint of the ALS-FTD continuum.

We described herein a complete loss of communication and even of response to command (except in a transitional recording $T_1$ for Patient 1), as assessed with multiple paraclinical tests, including a BCI, in two ALS patients who both had demonstrated communication more than four years after the onset of tetraplegia and tracheostomy. Both lost behavioural functional communication less than one year before the first neurophysiological attempt to assess their cognitive residual abilities. Patient 1 demonstrated a neurophysiological response to simple command for at least three months after the last saccade-based behavioural communication. We decided to try a non-invasive BCI in order to restore communication. The auditory modality was proposed as oculomotor impairment is known to prevent the use of visual BCI[16,58,59]. But there were ultimately multiple arguments to support that the absence of control of the BCI was due to the intrinsic evolution of the causal disease, as previously suspected in the literature[56,60].

In a previous study[24], electro-corticography (ECoG) was used for four sessions of event-related potentials (semantic and auditory stimulus) during six months. The results remained stable before the last communication using motor contact. Three months after the patient fulfilled cLIS criteria, P300 responses were no longer detected, concomitantly with a persisting MMN and an increased power of low frequency bands. This last neurological state was not discussed as an end-stage ALS or dementia but rather as the consequence of an intercurrent infection. Similarly, another study focusing on semantic cognitive process in a single patient with a cLIS found a progressive loss of responses to oddball and semantic paradigm but neither classic MMN nor imaging was performed to further describe this phenotype[61]. More generally, several electrophysiological features seem to be altered in advanced stages of ALS[54,62,63].

Neurophysiological signs of awareness were still observed three months after the last communication of Patient 1 ($T_1$), as far as clear effects in the "Count" and "Focused *vs* Diverted" conditions were observed in the active auditory oddball paradigm. This patient might have been less cognitively impaired at $T_1$ (before a final cognitive worsening at $T_2$ when the SON-P300 disappeared as well) than the one from in the previous ECoG study[24] and Patient 2 who presented an inconsistent response to the Act-Pass paradigm at his unique MAAB. One could argue that each of these cLIS candidates presented a particular dynamic of a common cognitive deterioration. In this hypothesis, the contingent dates of neurophysiological assessments would provide few screenshots of a shared continuum, ranging from a transient cLIS with attentional disorders (Patient 1 at $T_1$) to an MCS-like status with difficulties to answer simple commands (Patient 2) and ultimately a more severe type of DOC, mimicking an Unresponsive Wakefulness Syndrome with abnormal EEG background rhythm associated to the complete loss of command following (Patient 1 at $T_2$ and other unresponsive patients from the litterature[24]).

Assuming that awareness can only be demonstrated by its presence when a reportable communication is confirmed, isolated negative BCI results were a necessary but not sufficient proof of awareness impairment. As the behavioural/motor gold standard was unavailable to confirm the presence or absence of awareness, we reinforced the methodological confidence by using alternative methods. Eventually, the MAAB was performed three times to test the following null hypothesis "the patient has no sign of awareness" using complementary levels of analysis. For Patient 1, we were initially able to reject this hypothesis in June of Year 5 ($T_1$, before BCI testing) thanks to the patient's ability to consistently follow a simple command during the Act-Pass session. For Patient 1 at $T_2$ and for Patient 2, three consecutive negative tests (failed auditory BCI, absence of consistent responses to the Act-Pass paradigm, and failed activation-task imaging) did not allow to reject the null hypothesis. In other words, the presence of any reportable consciousness could not be proven any more at the individual level. At the general and theoretical level, it implied that we rejected the hypothesis stating that "end-stage ALS patients have only a behavioural communication issue" because DOC can occur at a final stage.

As neither neurophysiological nor imaging methods have been validated to diagnose the absence of awareness, the proposal of absence of awareness remained hypothetical and had to be cross-validated by complementary arguments confirming the impairment of basal brain functions supporting the global state of consciousness[64,65].

Regarding the wakefulness dimension of consciousness[65], EEG background activity for Patient 1 at $T_1$ was compatible with a normal wake pattern, as previously described in typical cLIS patients[33]. In line with previous results[24], a progressive slowing of EEG background rhythms was observed for Patient 1 at $T_2$ and for Patient 2 during day-time EEG. Long-term EEG monitoring ruled out the hypothesis that wakefulness fluctuations (with sleep occurring unexpectedly during the day) could be solely responsible for the loss of all response to passive and active auditory paradigm: the sleep/wake architecture was abnormal but N2 and N3 occurred predominantly during the night for both patients and the "pseudo-wake" patterns predominating during the day were abnormal as well. Altogether, the EEG patterns were compatible with the continuum between sleep-like

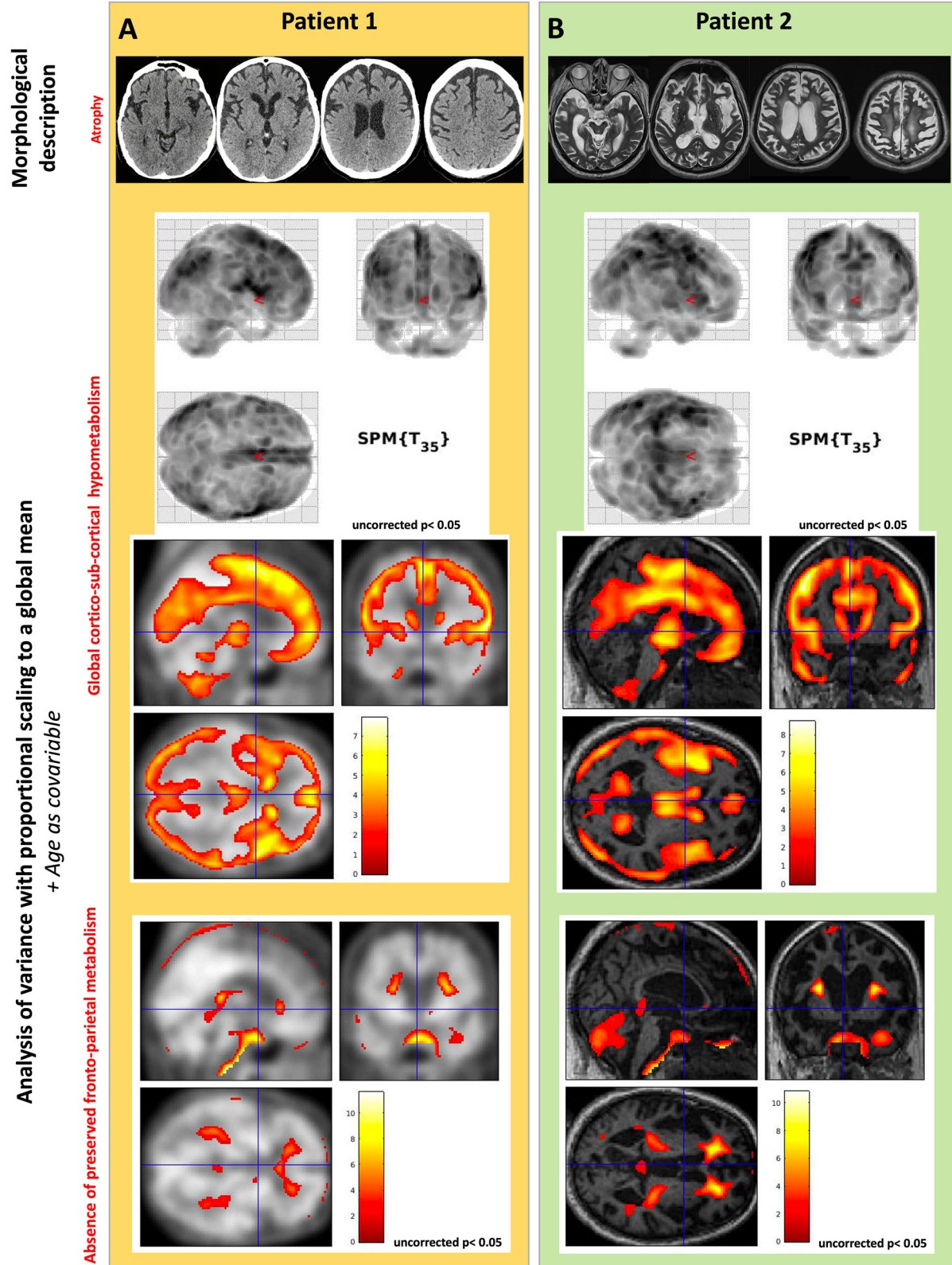

**Fig. 5 | Morphological and metabolic imaging. A** Patient 1: For PET illustration, the results are projected on the mean PET image of healthy subjects (in the absence of MRI). **B** Patient 2: For PET illustration, the results are projected the simultaneously acquired MRI. Top part: Focal lobar frontal and temporal atrophy (CT scan for Patient 1) and diffuse atrophy (MRI scan for Patient 2). Middle part: [18F]FDG-PET contrast "Patient <Healthy subjects" using an analysis of variance (on SUV images, uncorrected, $p < 0.05$) demonstrating global hypometabolism, in particular for the associative fronto-parietal cortex. Bottom part: [18F]FDG -PET contrast "Patient > Healthy subjects" using an analysis of variance (on SUV images, uncorrected, $p < 0.05$) demonstrating global hypermetabolism. Colour bars: t-scores. All illustrations are shown for the MNI coordinates (X = −1.45, Y = 3, Z = −0.68).

and a pathological (coma-like) state of wakefulness[66] but as EEG was not continuously recorded during every passive or activation task, unexpected sleep or sleepiness period could not be formally ruled out.

Concerning the assessment of cortical function required for consciousness[67], the deteriorated neurophysiological responses were compatible with an altered processing of cognitive information[68]. An altered signal/noise ratio due to muscle artefacts was highly unlikely in ALS, given the depressed muscular tone. Recent EEG data confirmed the impairment of the EEG spectrum (shift towards slow waves at rest) in cLIS-ALS patients[34], that was not related to an awareness issue by the authors.

Metabolic data gave strong arguments to integrate morphological and functional information into a reliable framework. In a comparative study between MRI voxel-based morphometry and FDG-PET in patients with ALS-FTD[69], the hypometabolism was more widespread than the decrease of cortical thickness and better correlated to the clinical features. This earlier occurrence of decreased metabolism before the appearance of atrophy on morphological imaging may be explained by the relationship between the functional FDG-PET imaging and the synaptic activity itself. FDG is metabolised in mitochondria at the synaptic level and its uptake is correlated to the synaptophysin level, a marker of synaptic density[70]. Despite this physiological argument, confirming whether the metabolic impairment anticipates the extension of atrophy in ALS would require rigorous longitudinal studies.

Extrapolating from the global extension of hypometabolism and the related clinical severity[2], we argue that the present snapshot of ALS evolution describes a further step in ALS pathophysiology. Indeed, the extent of the "full-blown FTD"[2] hypometabolism was not comparable to the one observed in the fronto-parietal areas[45,71] among the present end-stage ALS patients. In our cases, it was more in line with Unresponsive Wakefulness Syndrome patients, the most severe expression of DOC. The involvement of extra-frontal areas could explain how a limited cognitive dysfunction evolves towards consciousness impairment, in line with both the global neuronal workspace hypothesis[72] (fronto-parietal functional impairment) and the mesocircuit hypothesis[73] (basal ganglia functional impairment).

Altogether, and contrary to other authors maintaining a segregation between the cLIS and DOC litteratures[32,34,35], we hypothesise that late-stage cLIS-ALS can evolve through a FTD-ALS continuum to a progressive extinction of consciousness related to a multisystemic central involvement. It impairs wakefulness (as in the present polysomnographic results and in ref. [33]), spectral and complexity EEG prerequisites of awareness (as in the present EEG results and in refs. [32,34]), command-following in different paradigms such as active oddball paradigm, activation task imaging and BCI adapted to the auditory stimulus (as in the present attempt and in refs. [30,32,35]), robust primary cortex responses (as in the present abnormal SEPs response with a reduced amplitude of the cortical response for Patient 2 and in ref. [35]) and resting metabolism (as in the present unique demonstration). Two mechanistic hypotheses can be proposed to explain this final ALS stage.

First, a progressive "extinction of thought"[16] can be discussed. It would be primarily due to the loss of afferent stimulus during cLIS, explaining why this extreme stage was proposed as an exclusion criterion to test a reliable communication device, based on a meta-analysis in various aetiologies (but mostly in ALS)[56]. In this hypothesis, the global hypometabolism and the subsequent atrophy would be a consequence of the cLIS, eventually caught out by a subsequent inability for any motor mental imaging.

Second, the final stage of the ALS neurodegenerative process could be compatible with a dynamic of cognitive decline towards DOC. This hypothesis is in line with the pathological features[19] of cLIS showing the impairment of the mesocircuit[73] and of the brainstem reticular formation[25]. This evolution could also be compared to other causes of neurodegeneration, such as Creutzfeldt-Jakob disease [CJD] in case reports[74,75] or small series[76]. Despite the classically recognised existence of neuro-cognitive impairment in ALS[77], the respective speed of severe cognitive dysfunction (fast in CJD and slow in ALS) and respiratory failure (slow or absent in CJD and fast in ALS) implies that this final stage would be more probably observed among CJD cases. The brain metabolism of ALS-FTD might

reach, after years a functional level comparable to the one observed a few months after CJD onset.

Of note, our present description does not allow causal inference to disentangle these mechanisms.

This description had several limitations. The main limitation is the small number of patients that could be included in this observatory study, due to the extreme rarity of such cases in a single ALS centre, particularly in France, where most patients declined artificial ventilation, and by the necessity to avoid moving patients from other centres, which would have provided them access to our complete awareness evaluation.

From a methodological standpoint, the analyses did not include systematic comparisons with normative datasets that specifically contrast the two patients with a cohort of healthy controls. Furthermore, the proposed degenerative DOC profile was not directly compared to more common aetiologies of DOC, such as acquired acute brain injuries (e.g., stroke, anoxic encephalopathy, traumatic brain injury), which could have provided further contextualization of the findings. However, several of the neurophysiological metrics have already been contextualized with normative data or previous studies involving non-degenerative DOC populations. The Act-Pass Paradigm was previously tested in healthy individuals ($n = 20$) as described in ref. [38] and DOC patients ($n = 68$) as described in ref. [78]. BCI Paradigm was previously tested in 18 healthy individuals and 3 non-ALS patients as described in ref. [41]. Concerning the EEG Analysis, the result includes an illustrative example of a healthy subject. The ASEEGA algorithm was trained on normative data as described in ref. [36]. For PET imaging, the provided analysis is explicitly based on a control group of 37 healthy subjects previously described in ref. [46]. Finally, mental imagery task analysis (PET and fMRI) are based on contrasts performed within subjects and without external control data[27,47], despite previously validated among a population of healthy subjects[27].

Therefore, these results should be confirmed in further studies and possibly with other forms of neurodegenerative diseases. Moreover, the pattern of diffuse hypometabolism among degenerative DOC could be compared to a PET database including conscious ALS patients, ALS-FTD patients, FTD patients and patients presenting a post-ABI DOC following different causes of acute brain injuries (anoxic encephalopathy, traumatic brain injury, stroke) as they usually have such comprehensive brain function assessment during the neuro-prognostication process (prolonged EEG[65,79], passive odd-ball paradigm[80,81], active paradigm of brain activation detection to assess Cognitive Motor Dissociation[26,78], morphological imaging[82] and functional neuro-imaging[45,83]). Developing such complementary cohorts would allow formal comparison between each mechanism of DOC (post-ABI and degenerative) as well as the specific degenerative patterns of each natural history (previous cognitive–ALS–FTD, or purely motor degeneration – ALS) so as to formalise specific pathways. The natural trajectory for the primary motor dysfunction might be: tetraplegia → LIS → cLIS – possibly very transient → DOC – non mandatory (with a FTD occurring possibly at any step). On the contrary, one could argue that the natural trajectory of a primary cognitive dysfunction might be: FTD → tetraplegia → DOC – highly probable.

From a public health perspective, one could argue that this study does not establish the prevalence of such extreme conditions, as it is based on rare case reports. Indeed, its primary objective was to expand the nosographical definition of end-stage ALS to include the dimension of disorders of consciousness (DOC). However, this should not deter physicians, patients, and families from addressing communication management in cases of prolonged ALS. While this outcome is by no means inevitable, the findings of the present study may help inform patients most directly affected of all potential trajectories when tracheostomy is considered. As such, the possibility of having developed a neurodegenerative DOC should be discussed when behavioural modes of communication have failed. Finally, the condition may serve as a plausible explanation in cases where paraclinical assessments do not yield conclusive results concerning the ability to manipulate a BCI model. Nonetheless, this does not preclude the use of advanced BCIs, especially given the promising complementary findings

supporting their role in sustaining communication in patients with tetra-plegic and anarthric ALS[31] so as to maintain their quality of life[84].

In the future, a longitudinal study should describe more ALS patients after the loss of oral communication and compare them to cLIS from different causes and to severe FTD without ALS. However, managing such a study seems difficult due to logistical issues and the rarity of such cases, their occurrence being the combination of ALS incidence and of the willingness to survive in a cLIS condition.

Such a study would require a major scientific effort in implementing: i) multiple types of assessment to reproduce the MAAB; ii) complementary BCI to test their respective sensitivity; iii) in a large cohort at a multi-centric international scale; iv) with its proper financial and societal issues as the cultural representation of care continuation in end-stage ALS is highly heterogeneous, even among high-income countries.

Despite being perfectly justified, its feasibility within a multi-centric consortium is not easy to assure in real-world clinical research. It would be more plausible to expand the analyses in two complementary directions.

First of all, a large-scale prevalence assessment with non-behaviourally communicating patients who are susceptible to having a DOC-like profile. It would include in a large number of centres to perform a non-invasive, consensual and acceptable BCI test for communication testing, together with a prolonged EEG for wakefulness evaluation, a morphological imaging to assess atrophy and a functional imaging to confirm diffuse hypometabolism. The respective prevalence of cLIS and degenerative DOC would then be clearly deciphered. For some patients presenting a confirmed cLIS profile, the occurrence and, if required, the delay for DOC transition would be assessed in a longitudinal ancillary analysis.

In a secondary in-depth study, highly motivated ALS patients and relatives could be recruited when they are communicating with eyes only, but are at risk of evolving towards cLIS or the degenerative DOC profile. The study would be longitudinal (with the same extensive battery repeated several times, e.g. twice a year) and would compare at each point the sensitivity and accuracy of communicative response provided by a large (despite non exhaustive) battery of test (non-invasively and possibly invasively), whose middle-term and short-term repetitions would be used to assess test-retest reliability and patients' fatigability, respectively.

## Conclusion

The comprehensive description of these two cases suggests that, for patients after a prolonged period of mechanical ventilation allowing survival beyond the natural history of ALS, the extreme evolution of an associated ALS-FTD pattern could induce a complete loss of communication, which could resist to current paraclinical tools implemented to overcome the motor failure. These patients might have reached a DOC stage instead of a functional LIS[23]. Thanks to a coherent corpus of arguments based on neurophysiological and imaging approaches, we proposed that such cases match more accurately the description of a "degenerative DOC". This point could have important psychological and ethical consequences for patients entering the final phase of ALS: waiting for BCI optimisation and success remains legitimate, but this perspective appears as equivocal. Informing patients and caregivers that consciousness itself could be impaired at last is of utmost interest to plan state-of-the-art advance directives.

Ethical discussions in DOC after an acute brain injury usually face several uncertainties. In absence of an explicit written patients' will, what level of disability would have they considered as acceptable? The time course of ALS allows to address these questions in advance, when the patient maintains communication abilities. Another difference between ALS and classic post-injury DOC is the dynamic of the disease (negative by degenerative processes vs plausibly positive by plasticity). Despite the delay between steps may vary (the cLIS could occur between 3 years since diagnosis as in ref. 55, to 5 years for Patient 1 herein or 15 years for the patient in ref. 30), the comparable progressive loss of communication illustrated here and by other studies[30,55] reduces the uncertainty concerning the prognosis. Once a cLIS is observed and consciousness impairment is confirmed using a dedicated battery, no return to communication should be expected as no curative treatment exists so far. In future cases, this dramatic cognitive decline should be anticipated before communication disappears to enable precise advance directives regarding end-of-life issues in case complete – and neurophysiologically confirmed – unresponsiveness occurs.

More theoretically, this study illustrates how much the diagnosis of disorders of consciousness is a conceptual challenge in the absence of any behavioural clue. This challenge could be overcome by convergent para-clinical arguments demonstrating the absence of awareness, as shown here. More importantly, the response can be reinforced by the absence of major prerequisites of awareness, such as physiological changes of wakefulness and a maintained resting metabolism.

## Data availability
Anonymized imaging and neurophysiological data (excluding clinical data) are available from the corresponding author upon reasonable request. No public data repository has been used. As the figures presented are purely illustrative and do not contain numerical graphs, no source data are available.

## Code availability
The code used for imaging data analysis (excluding BCI-related code) is also available from the corresponding author upon reasonable request.

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

## Author contributions

Clinical data: F.G., F.D. and E.B. performed the A.L.S. recruitment and clinical observations. Neurophysiology: E.M., D.M. and J.M. conceived and designed the brain-computer interface and the experimental paradigm. E.M., J.M. and A.O. designed and implemented the brain-computer interface. E.M., A.O. and J.M. performed the BCI tests. P.S., E.M. and J.M. performed the offline data analysis. C.B. conceived and designed the hypnogram algorithm and performed the sleep data analysis. Act-Pass: D.M. and N.A.O. conceived and designed the paradigm. E.M., J.J., J.M. and R.B. designed and implemented the current version of the paradigm. E.M., A.O. and J.M. performed the Act-Pass tests. J.J. and R.B. performed the offline data analysis. Imaging: F.G., C.S., D.L, N.C. and I.M. performed the PET acquisitions. A.H., N.C and I.M. performed the PET data analysis.

## Competing interests

The authors declare the following competing interests AO was funded by the Fondation pour la Recherche Médicale (FRM, ING20121226307). PS, JM, EM were funded by one grant from the Fondation pour la Recherche Médicale (FRM, FDM201906008524). JM, EM and PS were funded by ANR-17-CE40-0005, MindMadeClear & ANR-20-CE17-0023, ANR HiFi. PS was funded by Perce-Neige Fondation. The teams of the Lyon Neurocience Research Centre are funded by the Labex cortex. All other authors declare no competing interests.

## Additional information

**Peer review information** : *Communications Medicine* thanks Calixto MacHado and the other, anonymous, reviewer(s) for their contribution to the peer review of this work. [A peer review file is available].

[1]Hospices Civils de Lyon, Neurological hospital Pierre-Wertheimer, Intensive Care Unit, Lyon, France. [2]Lyon Neuroscience Research Center, CAP Team, Lyon, France. [3]CERMEP - Imagerie du vivant, Lyon, France. [4]Lyon Neuroscience Research Center, COPHY Team, Lyon, France. [5]Hospices Civils de Lyon, Neurological Hospital Pierre-Wertheimer, Functional Neurology and Epileptology unit, Lyon, France. [6]PHYSIP SA, Paris, France. [7]Institut de Chimie et de Biochimie Moléculaires et Supra-moléculaires, Lyon, France. [8]Hospices Civils de Lyon, Cardiological hospital Louis Pradel, Nuclear medicine unit, Lyon, France. [9]Neurodis Foundation, Lyon, France. [10]King's College London & Guy's and St Thomas' PET Centre, School of Biomedical Engineering and Imaging Sciences, King's College London, London, UK. [11]Hospices Civils de Lyon, Neurological Hospital Pierre-Wertheimer, Lyon, France. [12]These authors contributed equally: Florent Gobert, Inès Merida, Emmanuel Maby. ✉e-mail: florent.gobert01@chu-lyon.fr

