## [Transparent Peer Review file · Communications Medicine]

Disorder of consciousness rather than complete Locked-In Syndrome for end stage Amyotrophic Lateral Sclerosis: A case series

Corresponding Author: Dr Florent Gobert

Version 1:

Reviewer comments:

Reviewer #1

(Remarks to the Author)

This paper presents a case report wherein the authors evaluated the level of alertness in two patients with amyotrophic lateral sclerosis (ALS) and complete locked-in syndrome (cLIS) who underwent tracheostomy positive pressure ventilation (TPPV). They attempted to evaluate the cerebral function radiologically and physiologically in patients with ALS–cLIS. Although the number of patients was small (two patients), this study succeeded in providing clinically useful and suggestive information about the cerebral function of patients with ALS in the terminal stage. The main text has been carefully written and well-structured.

The authors concluded that the terminal features of cerebral lesions in ALS do not lead to LIS, in which consciousness cannot be expressed, but rather to a more general decline in alertness due to cerebral damage. A subgroup of patients with ALS supported this conclusion. However, clear differences were observed in the progression of ALS-related cerebral lesions depending on the case, and not all patients reach cLIS uniformly within a few years. In other words, while some patients develop a condition close to cLIS within a few years, a certain number of patients survive under artificial respiration for several to a dozen years, becoming completely quadriplegic but not reaching cLIS. This should be clearly stated in the Discussion. In neurodegenerative diseases, including ALS, the brain atrophy progresses over time and many patients are unable to communicate. The authors should clearly state in their paper that, for at least some patients with ALS, the period before reaching cLIS is prolonged, and the use of a sophisticated brain–computer interface (BCI) during that period is likely to improve their quality of life (QOL).

Reviewer #2

(Remarks to the Author)

Review of the Manuscript on End-Stage ALS and Consciousness Assessment

This study addresses a crucial and controversial aspect of end-stage amyotrophic lateral sclerosis (ALS): whether consciousness is preserved in patients presumed to be in a complete Locked-In Syndrome (cLIS) or whether they instead transition into a state resembling a degenerative disorder of consciousness (DOC). The authors shift the conceptual framework from assuming preserved consciousness in cLIS to investigating Cognitive Motor Dissociation (CMD), where consciousness remains to be demonstrated.

Strengths of the Study

Multimodal Approach: The authors employ a robust battery of neurophysiological assessments, EEG-based wakefulness monitoring, passive and active auditory oddball paradigms, an auditory Brain-Computer Interface (BCI), and activation-task imaging. This comprehensive methodology strengthens the study's ability to probe the presence or absence of consciousness.

Objective Measures: The integration of functional neuroimaging (fMRI and PET) to assess brain metabolism and atrophy provides valuable insights into the neural mechanisms underlying end-stage ALS. The identification of widespread cortico-subcortical hypometabolism consistent with frontotemporal dementia (FTD) suggests a pathological process beyond pure

motor degeneration.

Clinical Significance: By challenging the assumption that cLIS represents an intact consciousness that is merely inaccessible, this study has implications for patient care, end-of-life decisions, and ethical considerations surrounding communication and autonomy in ALS patients.

Limitations and Areas for Improvement

Sample Size and Generalizability: The study is limited to two patients, which, while understandable given the rarity of the condition, limits the generalizability of the findings. Future studies should aim for larger cohorts to confirm whether the observed neurophysiological and metabolic changes are consistent across end-stage ALS cases.

Interpretation of Negative Findings: The failure of both patients to respond to the BCI and auditory oddball paradigms is presented as evidence of an underlying DOC. However, it remains unclear whether technical limitations, motor fatigue, or fluctuating attention states could have influenced these outcomes. Additional trials or alternative BCI modalities could help differentiate between residual awareness and true cognitive impairment.

Longitudinal Tracking: While Patient 1 showed an initial ability to follow commands before losing this ability, a more detailed timeline of cognitive decline could help delineate the trajectory from cLIS to a DOC-like state. Serial imaging and electrophysiological tracking could provide a clearer picture of when and how this transition occurs.

Need for Control Groups: Comparing ALS-cLIS patients with known DOC cases (e.g., patients with anoxic brain injury or FTD without ALS) would help distinguish ALS-specific degeneration from general mechanisms of consciousness loss.

Conclusion and Future Directions

This study presents compelling evidence that the final stage of ALS may not always conform to the classical notion of cLIS, but rather, in some cases, may resemble a degenerative disorder of consciousness. The neuroimaging findings significantly contribute to our understanding of the mechanisms at play in end-stage ALS and highlight the need for a paradigm shift in how we approach these patients.

Yes, it is necessary to continue further research to confirm these findings. Expanding this work with larger sample sizes, longitudinal monitoring, and refined neurophysiological and BCI methodologies will be essential to clarify the prevalence and implications of this proposed DOC-like stage in ALS.

Version 2:

Reviewer comments:

Reviewer #1

(Remarks to the Author)

The satisfying modification were made.

We would like to express again our sincere gratitude to the editorial team of *Communications Medicine* for their support and for granting our appeal following the initial editorial rejection. We are also truly thankful to the editors for coordinating the review of our manuscript and to the reviewers for their insightful and constructive comments.

By carefully addressing the reviewers' suggestions and thoroughly revising the manuscript, we believe that its quality has been significantly enhanced. Specifically, we revised the Abstract and the Discussion section extensively to: i) indicate to which extend a comparison of the two patients with normative databases was already available from literature, as far as possible; ii) clarify the limited implications of our findings for patients in the late stages of ALS who continue to use BCI for communication in cases of severe tetraplegia and classical (but not complete) Locked-In Syndrome.

Our point-by-point responses to the reviewers' comments are detailed below. The changes made in the revised manuscript (highlighted in the Article document) are marked in red thereafter.

Reviewer #1 (Remarks to the Author):

This paper presents a case report wherein the authors evaluated the level of alertness in two patients with amyotrophic lateral sclerosis (ALS) and complete locked-in syndrome (cLIS) who underwent tracheostomy positive pressure ventilation (TPPV). They attempted to evaluate the cerebral function radiologically and physiologically in patients with ALS–cLIS. Although the number of patients was small (two patients), this study succeeded in providing clinically useful and suggestive information about the cerebral function of patients with ALS in the terminal stage. The main text has been carefully written and well-structured.

The authors concluded that the terminal features of cerebral lesions in ALS do not lead to LIS, in which consciousness cannot be expressed, but rather to a more general decline in alertness due to cerebral damage. A subgroup of patients with ALS supported this conclusion. However, clear differences were observed in the progression of ALS-related cerebral lesions depending on the case, and not all patients reach cLIS uniformly within a few years. In other words, while some patients develop a condition close to cLIS within a few years, a certain number of patients survive under artificial respiration for several to a dozen years, becoming completely quadriplegic but not reaching cLIS.

This should be clearly stated in the Discussion. In neurodegenerative diseases, including ALS, the brain atrophy progresses over time and many patients are unable to communicate.

The authors should clearly state in their paper that, for at least some patients with ALS, the period before reaching cLIS is prolonged, and the use of a sophisticated brain–computer interface (BCI) during that period is likely to improve their quality of life (QOL).

We sincerely appreciate the reviewer's constructive comments and acknowledge this limitation in the implication of our study. To answer this issue, a paragraph of the Abstract has been changed as follow:

“This condition appears as a rare – but possible – endophenotype of ALS that could be patho-physiologically different from the classical tetraplegia and anarthria phenotype in which communication is behaviourally easily identifiable and for which the BCI manipulation can be preserved for long to increase communication understanding and speed so as to provide a better quality of life.”

A final paragraph of the Discussion has been modified as well to clearly state that our findings should not prevent physician, patient and family to manage communication in prolonged ALS: as no prevalence data are provided here, this final outcome is absolutely not certain but should just be considered as a possibility when tracheostomy is proposed, should be evoked when behavioural communication start failing, and can be used as a plausible explanation when no paraclinical tools are not proven successful.

“From a public health perspective, one could argue that this study does not establish the prevalence of such extreme conditions, as it is based on rare case reports. Indeed, its primary objective was to expand the nosographical definition of end-stage ALS to include the dimension of disorders of consciousness (DOC). However, this should not deter physicians, patients, and families from addressing communication management in cases of prolonged ALS. While this outcome is by no means inevitable, the findings of the present study may help inform patients most directly affected of all potential trajectories when tracheostomy is considered. As such, this possibility of having developed a neurodegenerative DOC should be discussed when behavioural modes of communication have failed. Finally, the condition may serve as a plausible explanation in cases where paraclinical assessments do not yield conclusive results concerning the ability to manipulate a BCI model. Nonetheless, this does not preclude the use of advanced BCIs, especially given the promising complementary findings supporting their role in sustaining communication in patients with tetraplegic and anarthric ALS¹ so as to maintain their quality of life².”

Reviewer #2 (Remarks to the Author):

Review of the Manuscript on End-Stage ALS and Consciousness Assessment

This study addresses a crucial and controversial aspect of end-stage amyotrophic lateral sclerosis (ALS): whether consciousness is preserved in patients presumed to be in a complete Locked-In Syndrome (cLIS) or whether they instead transition into a state resembling a degenerative disorder of consciousness (DOC). The authors shift the conceptual framework from assuming preserved consciousness in cLIS to investigating Cognitive Motor Dissociation (CMD), where consciousness remains to be demonstrated.

Strengths of the Study

Multimodal Approach: The authors employ a robust battery of neurophysiological assessments, EEG-based wakefulness monitoring, passive and active auditory oddball paradigms, an auditory Brain-Computer Interface (BCI), and activation-task imaging. This comprehensive methodology strengthens the study's ability to probe the presence or absence of consciousness.

Objective Measures: The integration of functional neuroimaging (fMRI and PET) to assess brain metabolism and atrophy provides valuable insights into the neural mechanisms underlying end-stage ALS. The identification of widespread cortico-subcortical hypometabolism consistent with frontotemporal dementia (FTD) suggests a pathological process beyond pure motor degeneration.

Clinical Significance: By challenging the assumption that cLIS represents an intact consciousness that is merely inaccessible, this study has implications for patient care, end-of-life decisions, and ethical considerations surrounding communication and autonomy in ALS patients.

We thank the reviewer for taking the time and effort to review our manuscript and for providing constructive comments.

Limitations and Areas for Improvement

Sample Size and Generalizability: The study is limited to two patients, which, while understandable given the rarity of the condition, limits the generalizability of the findings. Future studies should aim for larger cohorts to confirm whether the observed neurophysiological and metabolic changes are consistent across end-stage ALS cases.

We perfectly acknowledge for this limitation. In addition, we fully agree that it would not be manageable at the level of a single centre. However, one could argue that such “proof of concept” is necessary to convince different medical culture of the plausibility of this hypothesis before conducting such time-consuming and ethically-disputed research in a larger setting. This is also the reason why the first case report was not reported on its own, to show that such a

pattern was somehow reproducible, even in a consecutive patient and in a singular geographic area (Lyon and its surroundings).

A larger cohort however would only be made available by a multi-centric and plausibly international study with its proper financial and societal issues as the cultural representation of care continuation in end-stage ALS is highly heterogeneous, even among high-income countries, as exposed in the introduction: “Proposing and performing tracheostomy is highly dependent on the healthcare system and cultural background, varying from 5% in Europe to 30% in Japan.”

Interpretation of Negative Findings: The failure of both patients to respond to the BCI and auditory oddball paradigms is presented as evidence of an underlying DOC. However, it remains unclear whether technical limitations, motor fatigue, or fluctuating attention states could have influenced these outcomes. Additional trials or alternative BCI modalities could help differentiate between residual awareness and true cognitive impairment.

We are perfectly in accordance with this limitation, that we tried to theorize in the “*Absence of direct consciousness*” paragraph within the Discussion section.

We agree that, in case of failed communication, we can stand that “the MAAB [...] performed three times to test the following null hypothesis “the patient has no sign of awareness” using complementary levels of analysis” is not firmly sufficient to definitely reject the hypothesis of a present awareness if our technical means had a too low sensitivity to catch it.

This was the reason why we discussed explicitly this limit in:

“As neither neurophysiological nor imaging methods have been validated to diagnose the absence of awareness, the proposal of absence of awareness remained hypothetical and had to be cross-validated by complementary arguments confirming the impairment of basal brain functions supporting the global state of consciousness^{3,4}.”

In the necessary confirmative longitudinal analysis that should complete the evidence of this original findings, it would be required to include different BCI modalities (auditory, visual when authorised by residual oculo-motor abilities), complementary neurophysiological recordings (possibly invasive signal), additional technologies of signal computation and, if possible, confirmative technics using not only EEG but also imaging results.

Longitudinal Tracking: While Patient 1 showed an initial ability to follow commands before losing this ability, a more detailed timeline of cognitive decline could help delineate the trajectory from cLIS to a DOC-like state. Serial imaging and

electrophysiological tracking could provide a clearer picture of when and how this transition occurs.

We thank the reviewer for this comment which is in line with the attempt of longitudinal analysis for Patient 1 that was unfortunately not reproducible for Patient 2 who was less frequently transferred to the Lyon teaching hospital for logistical reason due to the duration of transport for a ventilated patient. Then, its cognitive status was only recorded while the communication was lost for a too long period to have a change to record the plausible transition from an authentic but short-lasting cLIS towards a degenerative DOC.

We perfectly agree that this probable natural history (tetraplegia → LIS → cLIS → DOC) could not be as linear as expected. It should be described in a proper longitudinal study. For example, the transition by a cLIS could not be systematical (some natural trajectories would be: tetraplegia → LIS → DOC). Moreover, as expressed by Reviewer 1, the second transition towards DOC might not be regarded as mandatory based on this sole illustrative description (some natural trajectories would be: tetraplegia → LIS → cLIS with endless communication abilities using the appropriate BCI modality and decoding algorithm).

Need for Control Groups: Comparing ALS-cLIS patients with known DOC cases (e.g., patients with anoxic brain injury or FTD without ALS) would help distinguish ALS-specific degeneration from general mechanisms of consciousness loss.

We appreciate the reviewer's apt and insightful comment.

Concerning the normative data based on healthy participants, the existence of such comparison is not directly provided in every result but most analysis have been previously validated in such a way.

Including several non-degenerative causes of acute brain injury [ABI] responsible for DOC (in an even more ambitious analysis concerning the general mechanism of awareness disappearance – either brutally or progressively) would be of utmost interest. This would nonetheless create even a greater organisational challenge as the medical pathways for each ABI mechanisms are distinct (acute DOC in intensive care units, ALS DOC in neurology department and in-home care). This is probably the reason why the different literatures (DOC and cLIS) were not interrogated so far in an integrated view. A more feasible approach would be to use in the end-stage ALS area some device and metrics that have been used previously to assess the awareness recovery in post-ABI DOC but are not considered as BCI. This is in particular the case for the machine-learning EEG detection of brain activation⁵ which has been

suggested to be theoretically equivalent to other methods of activation detection⁶, sometimes used as BCI⁷. Thus, incorporating DOC related metrics, using the CMD literature in the ALS area would help direct comparison between degenerative and post-ABI DOC. Of note, the active-passive paradigm already used in this study was initially developed for post-ABI DOC and the current results can be straightforwardly compared to previously published results⁸.

This point is now clearly stated in the discussion paragraph as follow:

“From a methodological standpoint, the analyses did not include systematic comparisons with normative datasets that specifically contrast the two patients with a cohort of healthy controls. Furthermore, the proposed degenerative DOC profile was not directly compared to more common aetiologies of DOC, such as acquired acute brain injuries (e.g., stroke, anoxic encephalopathy, traumatic brain injury), which could have provided further contextualization of the findings. However, several of the neurophysiological metrics have already been contextualized with normative data or previous studies involving non-degenerative DOC populations:

- Act-Pass Paradigm was previously tested in healthy individuals (n=20) as described by Morlet et al. in 2017⁹ and DOC patients (n=68) as described by Morlet et al. in 2023⁸.
- BCI Paradigm was previously tested in 18 healthy individuals and 3 non-ALS patients as described by Seguin et al. in 2024¹⁰
- EEG Analysis: the result includes an illustrative example of a healthy subject. The ASEEGA algorithm was trained on normative data as described by Berthomier et al., in 2007¹¹
- PET Imaging: the provided analysis is explicitly based on a control group of 37 healthy subjects previously described by Merida et al. in 2021¹²
- Activation analysis (PET and fMRI) are based on contrasts performed within subjects and without external control data^{7,13}, despite previously validated among a population of healthy subjects⁷.”

To indicate this future complementary perspective, a paragraph of the discussion section has been corrected as such:

Therefore, these results should be confirmed in further studies and possibly with other forms of neurodegenerative diseases. Moreover, the pattern of diffuse hypometabolism among degenerative DOC could be compared to a PET database including conscious ALS patients,

ALS-FTD patients, FTD patients and patients presenting a post-ABI DOC following different causes of acute brain injuries (anoxic encephalopathy, traumatic brain injury, stroke) as they usually have such comprehensive brain function assessment during the neuro-prognostication process (prolonged EEG^{4, 14}, passive odd-ball paradigm^{15, 16}, active paradigm of brain activation detection to assess Cognitive Motor Dissociation^{5, 8}, morphological imaging¹⁷ and functional neuro-imaging^{18, 19}). Developing such complementary cohorts would allow formal comparison between each mechanism of DOC (post-ABI and degenerative) as well as the specific degenerative patterns of each natural history (previous cognitive – ALS-FTD, or purely motor degeneration – ALS) so as to formalise specific pathways. The natural trajectory for the primary motor dysfunction might be: tetraplegia → LIS → cLIS – possibly very transient → DOC – non mandatory (with a FTD occurring possibly at any step). On the contrary, one could argue that the natural trajectory of a primary cognitive dysfunction might be: FTD → tetraplegia → DOC – highly probable.

Conclusion and Future Directions

This study presents compelling evidence that the final stage of ALS may not always conform to the classical notion of cLIS, but rather, in some cases, may resemble a degenerative disorder of consciousness.

The neuroimaging findings significantly contribute to our understanding of the mechanisms at play in end-stage ALS and highlight the need for a paradigm shift in how we approach these patients.

We appreciate the reviewer's insightful comment and hope that hybrid PET-MR imaging development will improve the availability of this multimodal approach to make it more feasible to assess brain function in relation to morphology in a larger setting.

Yes, it is necessary to continue further research to confirm these findings. Expanding this work with larger sample sizes, longitudinal monitoring, and refined neurophysiological and BCI methodologies will be essential to clarify the prevalence and implications of this proposed DOC-like stage in ALS.

We agree with the reviewer comment as stated in the following paragraph of the abstract which has been extended as follow.

Altogether, the neuroimaging features distinguishing the mechanisms in this rare condition is a significant milestone to understand end-stage ALS. The present clinical study calls for further longitudinal exploration in large scale population studies to determine the

prevalence of this degenerative disorder of consciousness profile among the end-stage ALS in whom communication seems hopeless.

Considering the issue which have been raised previously (“Interpretation of Negative Findings” with the necessity of using several complementary and alternative technics and “Longitudinal Tracking”) and the constraints related to the rarity of this condition (imposing a multi-centric design, as proposed in the “Sample Size and Generalizability”), a more comprehensive proposal appears. With the authorisation of editor, we propose to formally include this research design proposal in a dedicated paragraph of the discussion that would complete the current “**Limitations of the study**” paragraph in an extended “**Limitations of the study and perspectives for future research**”.

Such a study would require a major scientific effort in implementing: i) multiple types of assessment to reproduce the MAAB; ii) complementary BCI to test their respective sensitivity; iii) in a large cohort at a multi-centric international scale; iv) with its proper financial and societal issues as the cultural representation of care continuation in end-stage ALS is highly heterogeneous, even among high-income countries.

Despite perfectly justified, its feasibility within a multi-centric consortium is not easy to assure in real-world clinical research. It would be more plausible to expand the analyses in two complementary directions.

First of all, a large-scale prevalence assessment with non-behaviourally communicating patients who are susceptible to have a DOC-like profile. It would include in a large amount of centre to perform a non-invasive, consensual and acceptable BCI test for communication testing, together with a prolonged EEG for wakefulness evaluation, a morphological imaging to assess atrophy and a functional imaging to confirm diffuse hypometabolism. The respective prevalence of cLIS and degenerative DOC would then be clearly deciphered. For some patient presenting a confirmed cLIS profile, the occurrence and, if required, the delay for DOC transition would be assessed in a longitudinal ancillary analysis.

In a secondary in-depth study, highly motivated ALS patients and relatives could be recruited when they are communicating with eyes only but are at risk of evolving towards cLIS or the degenerative DOC profile. The study would be longitudinal (with the same extensive battery repeated several times, e.g. twice-a-year) and would compare at each point the sensitivity and accuracy of communicative response provided by a large (despite non exhaustive) battery of test (non-invasively and possibly invasively), whose middle-term and

short-term repetitions would be used to assess test-retest reliability and patients' fatigability, respectively.

1. Card NS, *et al.* An Accurate and Rapidly Calibrating Speech Neuroprosthesis. *N Engl J Med* **391**, 609-618 (2024).
2. Wolpaw JR, *et al.* Independent home use of a brain-computer interface by people with amyotrophic lateral sclerosis. *Neurology* **91**, e258-e267 (2018).
3. Bayne T, *et al.* Are There Levels of Consciousness? *Trends Cogn Sci* **20**, 405-413 (2016).
4. Gobert F, *et al.* Twenty-four-hour rhythmicities in disorders of consciousness are associated with a favourable outcome. *Commun Biol* **6**, 1213 (2023).
5. Claassen J, *et al.* Detection of Brain Activation in Unresponsive Patients with Acute Brain Injury. *N Engl J Med* **380**, 2497-2505 (2019).
6. Bodien YG, *et al.* Cognitive Motor Dissociation in Disorders of Consciousness. *N Engl J Med* **391**, 598-608 (2024).
7. Monti MM, *et al.* Willful modulation of brain activity in disorders of consciousness. *The New England journal of medicine* **362**, 579-589 (2010).
8. Morlet D, *et al.* Infraclinical detection of voluntary attention in coma and post-coma patients using electrophysiology. *Clin Neurophysiol* **145**, 151-161 (2023).
9. Morlet D, *et al.* The auditory oddball paradigm revised to improve bedside detection of consciousness in behaviorally unresponsive patients. *Psychophysiology* **54**, 1644-1662 (2017).
10. Seguin P, *et al.* The challenge of controlling an auditory BCI in the case of severe motor disability. *J Neuroeng Rehabil* **21**, 9 (2024).
11. Berthomier C, *et al.* Automatic analysis of single-channel sleep EEG: validation in healthy individuals. *Sleep* **30**, 1587-1595 (2007).
12. Merida I, *et al.* CERMEP-IDB-MRFXFDG: a database of 37 normal adult human brain [(18)F]FDG PET, T1 and FLAIR MRI, and CT images available for research. *EJNMMI research* **11**, 91 (2021).
13. Newey CR, *et al.* Optimizing SPECT SISCOM analysis to localize seizure-onset zone by using varying z scores. *Epilepsia* **54**, 793-800 (2013).
14. Cologan V, *et al.* Sleep in disorders of consciousness. *Sleep Medicine Reviews* **14**, 97-105 (2010).
15. Fischer C, *et al.* Predictive value of sensory and cognitive evoked potentials for awakening from coma. *Neurology* **63**, 669-673 (2004).
16. Fischer C, *et al.* Novelty P3 elicited by the subject's own name in comatose patients. *Clin Neurophysiol* **119**, 2224-2230 (2008).
17. Silva S, *et al.* Brain Gray Matter MRI Morphometry for Neuroprognostication After Cardiac Arrest. *Crit Care Med*, (2017).
18. Stender J, *et al.* Diagnostic precision of PET imaging and functional MRI in disorders of consciousness: a clinical validation study. *The Lancet* **384**, 514-522 (2014).
19. Di Perri C, *et al.* Neural correlates of consciousness in patients who have emerged from a minimally conscious state: a cross-sectional multimodal imaging study. *The Lancet Neurology*, (2016).